# Point-M2AE: Multi-scale Masked Autoencoders for Hierarchical Point Cloud Pre-training

**Renrui Zhang**[1,2]**, Ziyu Guo**[2]**, Rongyao Fang**[1]**,**
**Bin Zhao**[2]**, Dong Wang**[2]**, Yu Qiao**[2]**, Hongsheng Li**[1,3]**, Peng Gao**[✉2]

[1] CUHK-SenseTime Joint Laboratory, The Chinese University of Hong Kong,
[2] Shanghai AI Laboratory, [3] Centre for Perceptual and Interactive Intelligence Limited

{zhangrenrui, gaopeng}@pjlab.org.cn
hsli@ee.cuhk.edu.hk

## Abstract

Masked Autoencoders (MAE) have shown great potentials in self-supervised pre-training for language and 2D image transformers. However, it still remains an open question on how to exploit masked autoencoding for learning 3D representations of irregular point clouds. In this paper, we propose **Point-M2AE**, a strong **M**ulti-scale **MAE** pre-training framework for hierarchical self-supervised learning of 3D point clouds. Unlike the standard transformer in MAE, we modify the encoder and decoder into pyramid architectures to progressively model spatial geometries and capture both fine-grained and high-level semantics of 3D shapes. For the encoder that downsamples point tokens by stages, we design a multi-scale masking strategy to generate consistent visible regions across scales, and adopt a local spatial self-attention mechanism during fine-tuning to focus on neighboring patterns. By multi-scale token propagation, the lightweight decoder gradually upsamples point tokens with complementary skip connections from the encoder, which further promotes the reconstruction from a global-to-local perspective. Extensive experiments demonstrate the *state-of-the-art* performance of Point-M2AE for 3D representation learning. With a frozen encoder after pre-training, Point-M2AE achieves **92.9%** accuracy for linear SVM on ModelNet40, even surpassing some fully trained methods. By fine-tuning on downstream tasks, Point-M2AE achieves **86.43%** accuracy on ScanObjectNN, **+3.36%** to the second-best, and largely benefits the few-shot classification, part segmentation and 3D object detection with the hierarchical pre-training scheme. Code is available at https://github.com/ZrrSkywalker/Point-M2AE.

## 1 Introduction

Learning to represent from unlabeled data without annotations, known as self-supervised learning, has attained great success in natural language processing [10, 32, 33, 5], computer vision [19, 7, 8, 18] and multi-modality learning [31, 50, 21]. By pre-training on the large-scale raw data, the networks are endowed with robust representation abilities and can significantly benefit downstream tasks with fine-tuning. Motivated by masked language modeling [32, 10], MAE [18] and some other methods [46, 53, 3] adopt asymmetric encoder-decoder transformers [13] to apply masked autoencoding for self-supervised learning on 2D images. They represent the input image as multiple local patches, and randomly mask them with a high ratio to build the pretext task for reconstruction. Specifically, the encoder aims at capturing high-level latent representations from limited visible patches, and the lightweight decoder is forced to reconstruct the RGB values of masked patches on

36th Conference on Neural Information Processing Systems (NeurIPS 2022).

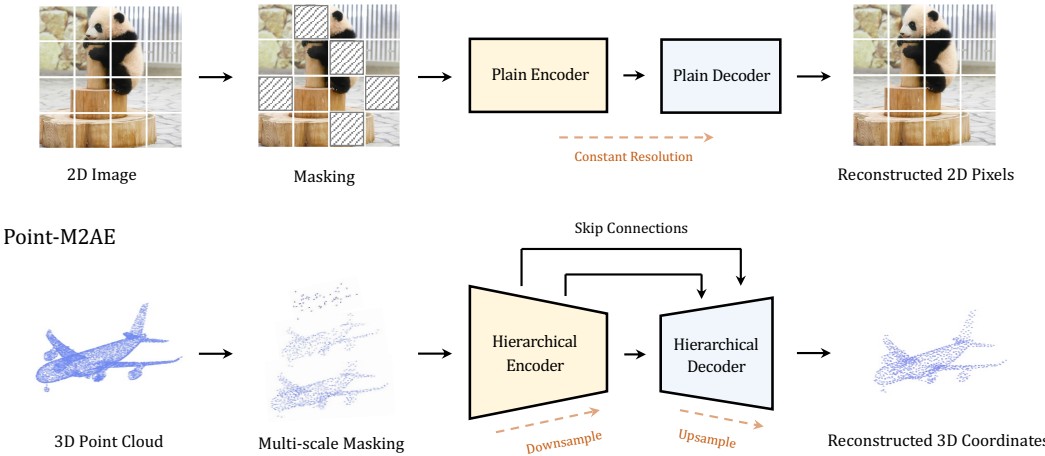

Figure 1: **Comparison of MAE (Top) and our Point-M2AE (Bottom).** MAE for 2D image pre-training adopts standard transformer of the plain encoder and decoder, while Point-M2AE introduces a hierarchical transformer with skip connections for multi-scale point cloud pre-training.

top. Despite its superiority on grid-based 2D images, we ask the question: can MAE-style masked autoencoding be adapted to irregular point clouds as a powerful 3D representation learner?

To tackle this challenge, we propose **M**ulti-scale **M**asked autoencoders for learning the hierarchical representations of point clouds via self-supervised pre-training, termed as Point-M2AE. We represent a point cloud as a set of point tokens depicting different spatial local regions, and inherit MAE's pipeline to first encode visible point tokens and then reconstruct the masked 3D coordinates. Different from 2D images, masked autoencoding for 3D point clouds has three characteristics to be specially considered. *Firstly,* it is critical to understand the relations between local parts and the overall 3D shapes, which have strong geometric and semantic dependence. As examples, the network can recognize an airplane starting from its wing, or segment the wing's part from the airplane's global feature. Therefore, we regard the standard transformer with the plain encoder and decoder is sub-optimal for capturing such local-global spatial relations in 3D, which directly downsamples the input into a low-resolution representation as shown in Figure 1 (Top). We modify both the encoder and decoder into multi-stage hierarchies for progressively encoding multi-scale features of point clouds, constructing an asymmetric U-Net [35] like architecture in Figure 1 (Bottom). *Secondly,* as our Point-M2AE encodes multi-scale point clouds unlike the single-scale 2D images, the unmasked visible regions are required to be both block-wise within one scale and consistent across scales, which are respectively for reserving complete local geometries and ensuring coherent feature learning for the network. For this, we introduce a multi-scale masking strategy, which generates random masks at the final scale with a high ratio (e.g., 80%), and back-projects the unmasked positions to all preceding scales. *Thirdly,* to better reconstruct 3D geometries from a local-to-global perspective, we utilize skip connections to complement the decoder with fine-grained information from the corresponding stages of the encoder. During fine-tuning on downstream tasks, we also adopt a local spatial self-attention mechanism with increasing attention scopes for point tokens at different stages of the encoder, which refocus each token within neighboring detailed structures.

By the multi-scale pre-training, Point-M2AE can encode point clouds from local-to-global hierarchies and then reconstructs the masked coordinates from global-to-local perspectives, which learns powerful 3D representations and performs superior transfer ability. After self-supervised pre-training on ShapeNet [6], Point-M2AE achieves 92.9% classification accuracy for linear SVM on ModelNet40 [44] with the frozen encoder, which surpasses the runner-up CrossPoint [2] by +1.2% and even outperforms some fully supervised methods. By fine-tuning on various downstream tasks, Point-M2AE achieves 86.43% (+3.36%) accuracy on ScanObjectNN [38] and 94.0% (+0.8%) accuracy on ModelNet40 [44] for shape classification, 86.51% (+0.91%) instance mIoU on ShapeNet-Part [48] for part segmentation, and 95.0% (+2.7%) accuracy on 10-way 20-shot ModelNet40 for

few-shot classification. Our multi-scale masked autoencoding also benefits the 3D object detection on ScanNetV2 [9] by +1.3% $AP_{25}$ and +1.3% $AP_{50}$, which provides the detection backbone with a hierarchical understanding of the point clouds.

We summarize the contributions of our paper as follows:

1. We propose Point-M2AE, a strong masked autoencoding framework, which conducts hierarchical point cloud encoding and reconstruction for better learning multi-scale spatial geometries of 3D shapes.

2. We introduce a U-Net like transformer architecture for MAE-style pre-training on point clouds, and adopt a multi-scale masking strategy to generate consistent visible regions across scales.

3. Point-M2AE achieves *state-of-the-art* performance for transfer learning on various downstream tasks, which indicates our approach to be a powerful representation learner for 3D point clouds.

## 2   Related Work

**Pre-training by Masked Modeling.**   Compared to contrastive learning methods [19, 7, 8] that learn from inter-sample relations, self-supervised pre-training by masked autoencoding builds the pretext tasks to predict the masked parts of the input signals. The series of GPT [32, 33, 5] and BERT [11] apply masked modeling to natural language processing and achieve extraordinary performance boost on downstream tasks with fine-tuning. Inspired by this, BEiT [4] proposes to match image patches with discrete tokens via dVAE [34] and pre-train a standard vision transformer [13, 49] by masked image modeling. On top of that, MAE [18] directly reconstructs the raw pixel values of masked tokens and performs great efficiency with a high mask ratio. The follow-up works further improve the performance of MAE by momentum encoder [53], contrastive learning [3], and modified reconstruction targets [42]. For self-supervised pre-training on 3D point clouds, the masked autoencoding has not been widely adopted. Similar to BEiT, Point-BERT [49] utilizes dVAE to map 3D patches to tokens for masked point modeling, but heavily relies on constrastive learning [19], complicated data augmentation, and the costly two-stage pre-training. In contrast, our Point-M2AE is a pure masked autoencoding method of one-stage pre-training, and follows MAE to reconstruct the input signals without dVAE mapping. Different from previous MAE methods adopting standard plain transformer, we propose a hierarchical transformer architecture along with the multi-scale masking strategy to better learn a strong and generic representation for 3D point clouds.

**Self-supervised Learning for Point Clouds.**   3D representation learning without annotations has been widely studied in recent years. Mainstream methods mainly build the pretext tasks to reconstruct the transformed input point cloud based on the encoded latent vectors, such as rotation [28], deformation [1], rearranged parts [36] and occlusion [40]. From another perspective, PointContrast [45] utilizes contrastive learning between features of the same points from different views to learn discriminative 3D representations. DepthContrast [51] further extends the contrast for depth maps of different augmentations. CrossPoint [2] conducts cross-modality contrastive learning between point clouds and their corresponding rendering images to acquire rich self-supervised signals. Point-BERT [49] and Point-MAE [27] respectively introduce BERT-style [10] and MAE-style [18] pre-training schemes for 3D point clouds with standard transformer networks and performs competitively on various downstream tasks, but both of them can only encode point clouds with a single resolution and ignores the local-global relations between 3D shapes. In this paper, we propose Point-M2AE, an MAE-style framework with a hierarchical transformer for multi-scale point cloud pre-training. We achieve *state-of-the-art* downstream performance by learning the multi-scale representation of point clouds.

## 3   Method

The overall pipeline of Point-M2AE is shown in Figure 2, where we encode and reconstruct the point cloud by a hierarchical network architecture. In Section 3.1, We first introduce the masking strategy of Point-M2AE with multi-scale representations of point clouds. Then in Section 3.2 and Section 3.3, we present the details of our encoder and decoder with multi-stage hierarchies.

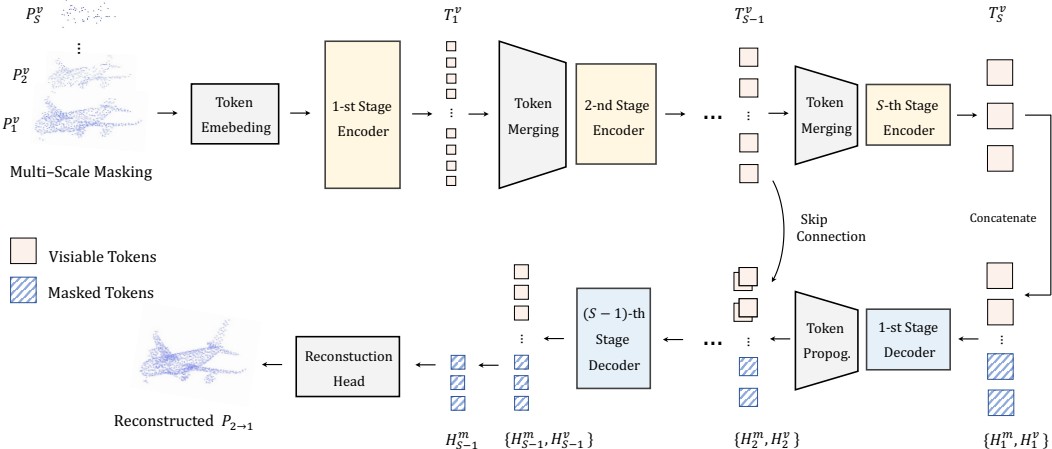

Figure 2: **Overall pipeline of Point-M2AE.** After the multi-scale masking, we embed point tokens at the 1-st scale and feed the visible ones into a hierarchical encoder-decoder transformer, which captures both high-level semantics and fine-grained patterns of the point cloud during pre-training.

## 3.1 Multi-scale Masking

To build a U-Net [35] like masked autoencoder for hierarchical learning, we encode the point cloud by $S$ scales with different number of points at each scale, and correspondingly modify the standard plain encoder into the $S$-stage architecture. Following MAE, we embed the point cloud into discrete point tokens and randomly mask them for reconstruction. Importantly, for irregular-distributed points in the multi-scale architecture, the unmasked visible spatial regions are required to be consistent not only within one scale, but also across different scales. This is because the block-wise parts of 3D shapes tend to preserve more complete fine-grained geometries, and the unmasked positions are better to be shared across all scales for coherent feature learning of the encoder. Therefore, as shown in Figure 3, we first construct the $S$-scale coordinate representations of the input point cloud and back-project the random masks from the final $S$-th scale to the earlier scales to avoid fragmented visible parts.

**$S$-scale Representations.**   We denote the input point cloud as $P \in \mathbb{R}^{N \times 3}$ and regard it as the 0-th scale. For the $i$-th scale, $1 \le i \le S$, we utilize Furthest Point Sampling (FPS) to downsample the points from the $(i-1)$-th scale, which produces seed points $P_i \in \mathbb{R}^{N_i \times 3}$ for scale $i$ of $N_i$ points. Then, we adopt $k$ Nearest-Neighbour ($k$-NN) to aggregate the neighboring $k$ points for each seed point and obtain the neighbor indices $I_i \in \mathbb{R}^{N_i \times k}$. By successively downsampling and grouping, we acquire the $S$-scale representations $\{P_i, I_i\}_{i=1}^S$ of the input point cloud, where the number of points $N_i$ gradually decreases and the inclusion relations between scales are recorded in $I_i$.

**Back-projecting Visible Positions.**   For seed points $P_S$ at the final $S$-th scale, we randomly mask them with a large proportion (e.g., 80%) and denote the remaining visible points as $P_S^v \in \mathbb{R}^{N_S^v \times 3}$ of $N_S$ points. We then back-project the unmasked positions $P_S^v$ to ensure the consistent visible regions across scales. For the $i$-th scale, $1 \le i < S$, we retrieve all the $k$ nearest neighbors of $P_{i+1}^v$ from the indices $I_{i+1}$ to serve as the visible positions $P_i^v$, and mask the others. By recursively back-projecting, we obtain the visible and masked positions of all $S$ scales, denoted as $\{P_i^v, P_i^m\}_{i=1}^S$, where $P_i^v \in \mathbb{R}^{N_i^v \times 3}$, $P_i^m \in \mathbb{R}^{N_i^m \times 3}$ and $N_i = N_i^v + N_i^m$.

## 3.2 Hierarchical Encoder

Based on the multi-scale masking, we embed the initial tokens of visible points $P_1^v$ for the 1-st scale and them into the hierarchical encoder with $S$ stages. Every stage is equipped with $K$ stacked encoder blocks, and each block contains a self-attention layer and a Feed Forward Network (FFN) of MLP layers. Between every two consecutive stages, we introduce spatial token merging modules to aggregate adjacent visible tokens and enlarge receptive fields for downsampling the point clouds.

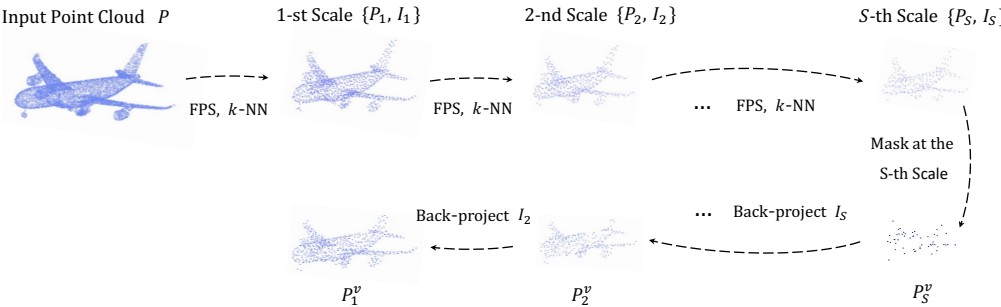

Figure 3: **Multi-scale masking strategy.** To obtain a consistent visible regions across scales, we first represent the input point cloud by multi-scale coordinates and generate the random mask at the highest one. Then, we back-project the unmasked visible positions to all earlier scales.

**Token Embedding and Merging.** Indexed by $I_1$, we utilize a mini-PointNet [29] to extract and fuse the features of every seed point from $P_1^v \in \mathbb{R}^{N_1^v \times 3}$ with its $k$ nearest neighbors. After that, we obtain the initial point tokens $T_1^v \in \mathbb{R}^{N_1^v \times C_1}$ for the 1-st stage of the encoder, which embeds $N_1^e$ local patterns of the 3D shape. Between the $(i-1)$-th and $i$-th stages, $1 < i \leq S$, we merge $T_{i-1}^v \in \mathbb{R}^{N_{i-1} \times C_{i-1}}$ to acquire the downsampled point tokens for the $i$-th stage. We utilize MLP layers and a max pooling to integrate every $k$ tokens nearest to $P_i^v$ indexed by $I_i$, which outputs $T_i^v \in \mathbb{R}^{N_i \times C_i}$. Due to our multi-scale masking, the merged $T_i^v$ corresponds to the same visible parts of $T_{i-1}^v$, which enables the consistent feature encoding across different scales. For larger $i$ of deeper stages, we set higher feature dimension $C_i$ to encode spatial geometries with richer semantics.

**Local Spatial Self-Attention.** During pre-training, we expect point tokens in the multi-stage encoder to capture global cues for 3D shapes, which benefits the reconstruction of masked parts. However, when fine-tuning on downstream tasks without masked autoencoding, point tokens in the shallower stages are better to mainly focus on local information and not to be disturbed by long-range signals, referring to the inductive bias of 3D locality [30]. Thus, during fine-tuning, we modify the original self-attention layer in the encoder with a local spatial constraint that only neighboring tokens within a ball query would be available for attention calculation. As the point tokens are downsampled by stages, we set increasing radii $\{r_i\}_{i=1}^S$ of multi-scale ball queries for gradually expanding the attention scopes, which fulfills the local-to-global feature aggregation scheme.

### 3.3 Hierarchical Decoder

Via the hierarchical encoder, we obtain the encoded visible tokens $\{T_i^v\}_{i=1}^S$ of all scales. Starting from the highest $S$-th scale, we assign a shared learnable mask token to all the masked positions $P_S^m$, and concatenate them with the visible tokens $T_S^v$. We denote them as $\{H_1^v, H_1^m\}$ with coordinates $\{P_S^v, P_S^m\}$, which serve as the input of the hierarchical decoder. We design the decoder to be lightweight with $S-1$ stages and only one decoder block for each stage, which enforces the encoder to embed more semantics of the point clouds. Each decoder block consists of a vanilla self-attention layer and an FFN. We do not apply the local constraint to the attention in the decoder, since a global understanding between visible and mask tokens is crucial to the reconstruction.

**Point Token Upsampling.** We upsample the point tokens between stages to progressively recover the fine-grained geometries of 3D shapes before reconstruction. We regulate that the $j$-th stage of the decoder corresponds to the $(S+1-j)$-th stage of the encoder, both of which contain point tokens of the same $(S+1-j)$-th scale with the feature dimension $C_{S+1-j}$. Between the $(j-1)$-th and $j$-th stage, $1 < j \leq S-1$, we upsample the tokens $\{H_{j-1}^v, H_{j-1}^m\}$ from the coordinates $\{P_{S+2-j}^v, P_{S+2-j}^m\}$ into $\{P_{S+1-j}^v, P_{S+1-j}^m\}$ via the token propagation module. Specifically, we obtain the $k$ nearest neighbors of each point token in $\{H_{j-1}^v, H_{j-1}^m\}$ indexed by $I_{S+2-j}$, and recover their neighbors' features by weighted interpolation referring to PointNet++ [30], which generates the tokens $\{H_j^v, H_j^m\}$ of the $j$-th stage.

Table 1: **Linear evaluation on Model-Net40 [44] by SVM.** We report different self-supervised learning methods and underline the second-best one.

| Method | Acc. (%) |
|---|---|
| 3D-GAN [43] | 83.3 |
| Latent-GAN [39] | 85.7 |
| SO-Net [22] | 87.3 |
| FoldingNet [47] | 88.4 |
| MAP-VAE [17] | 88.4 |
| VIP-GAN [16] | 90.2 |
| DGCNN + Jiasaw [37] | 90.6 |
| DGCNN + OcCo [40] | 90.7 |
| DGCNN + CrossPoint [2] | 91.2 |
| Transformer + OcCo [49] | 89.6 |
| Point-BERT [49] | 87.4 |
| **Point-M2AE** | **92.9** |
| *Improvement* | +1.7 |

Table 2: **Shape classification on ModelNet40 [44].** '#points' and 'Acc.' denote the number of points for training and the overall accuracy. [S] represents fine-tuning after self-supervised pre-training.

| Method | #points | Acc. (%) |
|---|---|---|
| PointNet [29] | 1k | 89.2 |
| PointNet++ [30] | 1k | 90.5 |
| PointCNN [23] | 1k | 92.2 |
| [S] SO-Net [22] | 5k | 92.5 |
| DGCNN [41] | 1k | 92.9 |
| PCT [15] | 1k | 93.2 |
| Point Transformer [52] | - | 93.7 |
| Transformer [49] | 1k | 91.4 |
| [S] Transformer + OcCo [49] | 1k | 92.1 |
| [S] Point-BERT [49] | 1k | 93.2 |
| [S] Point-BERT | 4k | 93.4 |
| [S] Point-BERT | 8k | 93.8 |
| **[S] Point-M2AE** | **1k** | **94.0** |

**Skip Connections.** To further complement the fine-grained geometries, we channel-wisely concatenate the visible tokens $H_j^v \in \mathbb{R}^{N_{S+1-j} \times C_{S+1-j}}$ of the decoder with $T_{S+1-j}^v \in \mathbb{R}^{N_{S+1-j} \times C_{S+1-j}}$ from the corresponding $(S+1-j)$-th stage of the encoder via skip connections, and adopt a linear projection layer to fuse their features. For the mask tokens $H_j^m$, we keep them unchanged, since the encoder only contains visible tokens without the masked ones.

**Point Reconstruction.** After $S-1$ stages of the decoder, we acquire $\{H_{S-1}^v, H_{S-1}^m\}$ with coordinates $\{P_2^v, P_2^m\}$ and reconstruct the masked values from the mask tokens $H_{S-1}^m$. Other than predicting values at the $0$-th scale of the input point cloud $P$, we reconstruct the coordinates of $P_1^m$, namely, recovering the masked positions of the $1$-st scale $P_1^m \in \mathbb{R}^{N_1^m \times 3}$ from the $2$-nd scale $P_2^m \in \mathbb{R}^{N_2^m \times 3}$. This is because $\{P_1^v, P_1^m\}$ of the $1$-st scale could well represent the overall 3D shape and simultaneously preserve enough local patterns, which already constructs a comparatively challenging pretext task for pre-training. If we further upsample $\{H_{S-1}^v, H_{S-1}^m\}$ into $\{H_S^v, H_S^m\}$ and reconstruct the masked raw points from $P_1^m$, the extra spatial noises and computational overhead would adversely influence our performance and efficiency. Therefore, for every token in $H_{S-1}^m \in \mathbb{R}^{N_2^m \times C_2}$, we reconstruct its $k$ nearest neighbors recorded in $I_2$ by a reconstruction head of one linear projection layer and compute the loss by $l_2$ Chamfer Distance [14], formulated as,

$$\widehat{P}_{2\to1}^m = \text{Linear}(H_{S-1}^m), \quad \text{where } \widehat{P}_{2\to1}^m \in \mathbb{R}^{N_2^m \times k \times 3}, \tag{1}$$

$$\mathcal{L}_{CD} = \text{ChamferDistance}(P_{2\to1}^m, \widehat{P}_{2\to1}^m), \tag{2}$$

where $\widehat{P}_{2\to1}^m$ and $P_{2\to1}^m$ denote the predicted and ground-truth reconstruction coordinates from the 2-nd scale to the 1-st scale. We only utilize $\mathcal{L}_{CD}$ for supervision without contrastive loss to conduct a pure masked autoencoding for self-supervised pre-training.

## 4 Experiments

In Section 4.1 and Section 4.2, we introduce the pre-training experiments of Point-M2AE and report the fine-tuning performance on various downstream tasks. We also conduct ablation studies in Section 4.3 to validate the effectiveness of our approach.

### 4.1 Self-supervised Pre-training

**Settings.** We pre-train our Point-M2AE on ShapeNet [6] dataset, which contains 57,448 synthetic 3D shapes of 55 categories. We set the stage number $S$ as 3, and construct a 3-stage encoder and a

Table 3: **Shape classification on ScanObjectNN [38]**. We report the accuracy (%) on the three splits of ScanObjectNN. [S] represents fine-tuning after self-supervised pre-training.

| Method | OBJ-BG | OBJ-ONLY | PB-T50-RS |
|---|---|---|---|
| PointNet [29] | 73.3 | 79.2 | 68.0 |
| PointNet++ [30] | 82.3 | 84.3 | 77.9 |
| DGCNN [41] | 82.8 | 86.2 | 78.1 |
| PointCNN [23] | 86.1 | 85.5 | 78.5 |
| Transformer [49] | 79.86 | 80.55 | 77.24 |
| [S] Transformer + OcCo [49] | 84.85 | 85.54 | 78.79 |
| [S] Point-BERT [49] | 87.43 | 88.12 | 83.07 |
| **[S] Point-M2AE** | **91.22** | **88.81** | **86.43** |
| *Improvement* | +3.79 | +0.69 | +3.36 |

2-stage decoder for hierarchical learning. We adopt 5 blocks in each encoder stage, but only 1 block per stage for the lightweight decoder. For the 3-scale point clouds, we set the point numbers and token dimensions respectively as {512, 256, 64} and {96, 192, 384}. We also set different $k$ for the $k$-NN at different scales, which are {16, 8, 8}. We mask the highest scale of point clouds with a high ratio of 80% and set 6 heads for all the attention modules. The detailed training settings are in Appendix.

**Linear SVM.** After pre-training on ShapeNet, we test the 3D representation capability of Point-M2AE via linear evaluation on ModelNet40 [44]. We sample 1,024 points from each 3D shape of ModelNet40 and utilize our frozen encoder to extract their features. On top of that, we train a linear SVM and report the classification accuracy in Table 1. As shown, Point-M2AE achieves the best performance among all existing self-supervised methods for point clouds, and surpasses the second-best CrossPoint [2] by +1.7%. Point-M2AE also exceeds Point-BERT [49] by +5.5%, which is a masked point modeling method with a MoCo loss [19] but adopts a standard transformer and conducts single-scale learning. It is worth noting that even if we freeze all our parameters, Point-M2AE with 92.9% accuracy still outperforms many fully trained methods on ModelNet40, e.g., 90.5% by PointNet++ [30], 92.8% by DensePoint [24], etc. The experiments fully demonstrate the superior 3D representation capacity of our Point-M2AE.

### 4.2 Downstream Tasks

For fine-tuning on downstream tasks, we discard the hierarchical decoder in pre-training and append different heads onto the hierarchical encoder for different tasks.

**Shape Classification.** We fine-tune Point-M2AE on two shape classification datasets: the widely adopted ModelNet40 [44] and the challenging ScanObjectNN [38]. For local spatial attention layers, we set the ball queries' radii of 3-scale point clouds as {0.32, 0.64, 1.28}. We follow Point-BERT to use the voting strategy [25] for fair comparison on ModelNet40. To handle the noisy spatial structures, we increase $k$ of $k$-NN into {32, 16, 16} for ScanObjectNN to encode local patterns with larger receptive fields. As reported in Table 2, Point-M2AE achieves 94.0% accuracy on ModelNet40 with 1024 points per sample, which surpasses Point-BERT fine-tuned with 1024 points by +0.8% and 8192 points by +0.2%. For ScanObjectNN in Table 3, our Point-M2AE outperforms the second-best Point-BERT by a significant margin, +3.79%, +0.69% and +3.36%, respectively for the three splits, indicating our great advantages under complex circumstances by multi-scale encoding. As ScanObjectNN of real-world scenes has a large semantic gap with the pre-trained synthetic ShapeNet, Point-M2AE also exerts strong transfer ability to understand point clouds of another domain.

**Part Segmentation.** We evaluate Point-M2AE for part segmentation on ShapeNetPart [48], which predicts per-point part labels and requires detailed understanding for local patterns. We adopt an extremely simple segmentation head to validate the effectiveness of our pre-training for well capturing both high-level semantics and fine-grained details. By the hierarchical encoder, we obtain

Table 4: **Few-shot classification on ModelNet40 [44]**. We report the average accuracy (%) and standard deviation (%) of 10 independent experiments.

| Method | 5-way | | 10-way | |
|---|---|---|---|---|
| | 10-shot | 20-shot | 10-shot | 20-shot |
| DGCNN [41] | $91.8 \pm 3.7$ | $93.4 \pm 3.2$ | $86.3 \pm 6.2$ | $90.9 \pm 5.1$ |
| [S] DGCNN + OcCo [40] | $91.9 \pm 3.3$ | $93.9 \pm 3.1$ | $86.4 \pm 5.4$ | $91.3 \pm 4.6$ |
| Transformer [49] | $87.8 \pm 5.2$ | $93.3 \pm 4.3$ | $84.6 \pm 5.5$ | $89.4 \pm 6.3$ |
| [S] Transformer + OcCo [49] | $94.0 \pm 3.6$ | $95.9 \pm 2.3$ | $89.4 \pm 5.1$ | $92.4 \pm 4.6$ |
| [S] Point-BERT [49] | $94.6 \pm 3.1$ | $96.3 \pm 2.7$ | $91.0 \pm 5.4$ | $92.7 \pm 5.1$ |
| **[S] Point-M2AE** | $\mathbf{96.8 \pm 1.8}$ | $\mathbf{98.3 \pm 1.4}$ | $\mathbf{92.3 \pm 4.5}$ | $\mathbf{95.0 \pm 3.0}$ |
| *Improvement* | +2.2 | +2.0 | +1.3 | +2.3 |

Table 5: **Part segmentation on ShapeNetPart [48]**. 'mIoU$_C$' (%) and 'mIoU$_I$' (%) denote the mean IoU across all part categories and all instances in the dataset, respectively.

| Method | mIoU$_C$ | mIoU$_I$ |
|---|---|---|
| PointNet [29] | 80.39 | 83.70 |
| PointNet++ [30] | 81.85 | 85.10 |
| DGCNN [41] | 82.33 | 85.20 |
| Transformer [49] | 83.42 | 85.10 |
| [S] Transformer + OcCo [49] | 83.42 | 85.10 |
| [S] Point-BERT [49] | 84.11 | 85.60 |
| **[S] Point-M2AE** | **84.86** | **86.51** |
| *Improvement* | +0.75 | +0.91 |

Table 6: **3D object detection on ScanNetV2 [9]**. We report the performance (%) of self-supervised learning methods based on VoteNet [12] and 3DETR-m [26].

| Method | AP$_{25}$ | AP$_{50}$ |
|---|---|---|
| VoteNet [12] | 58.6 | 33.5 |
| [S] STRL [20] | 59.5 | 38.4 |
| [S] PointContrast [45] | 59.2 | 38.0 |
| [S] DepthContrast [51] | 61.3 | – |
| 3DETR [26] | 62.1 | 37.9 |
| 3DETR-m [26] | 65.0 | 47.0 |
| **[S] Point-M2AE** | **66.3** | **48.3** |
| *Improvement* | +1.3 | +1.3 |

3-scale point tokens of {512, 256, 64} points, and perform feature propagation in PointNet++ [30] to independently upsample the tokens into 2048 points of the input point cloud. Then, we concatenate the upsampled 3-scale features for each point and predict the part label by stacked linear projection layers. As reported in Table 4.2, Point-M2AE achieves the best 86.51% instance mIoU with the simple segmentation head, surpassing the second-best Point-BERT by +0.91%. Note that Point-BERT [49] and other methods [29, 30, 41] adopt hierarchical segmentation heads to progressively upsample the point features from intermediate layers, while our head contains no hierarchical structure and only relies on the pre-trained encoder to capture the multi-scale information of point clouds. The results fully demonstrate the significance of Point-M2AE's multi-scale pre-training to segmentation tasks.

**Few-shot Classification.** We conduct experiments for few-shot classification on ModelNet40 [44] to evaluate the performance of Point-M2AE with limited fine-tuning data. As reported in Table 4.2, Point-M2AE achieves the best performance for all four settings, and surpasses Point-BERT by +2.2%, +2.0%, +1.3%, and +2.7%, respectively. Our approach also shows smaller deviations than other transformer-based methods, which indicates Point-M2AE has learned to produce more universal 3D representations for well adapting to downstream tasks under low-data regimes.

**3D Object Detection** To further evaluate our hierarchical pre-training on 3D object detection, we apply Point-M2AE to serving as the feature backbone on the indoor ScanNetV2 [9] dataset. We select 3DETR-m [26] as our baseline, which consists of a 3-block encoder and a transformer decoder. Considering the quite different dataset statistics, e.g., 2k input points for ShapeNet [6] and 50k input points for ScanNetV2, we adopt the same encoder architecture with that of 3DETR-m, and keep our hierarchical decoder with skip connections unchanged for self-supervised pre-training on ScanNetV2. More details of models and training are in Appendix. As reported in Table 4.2, compared to training from scratch, our hierarchical pre-training boosts the performance of 3DETR-m by +1.34% AP$_{25}$ and +1.29% AP$_{50}$. The experiments demonstrate the effectiveness of Point-M2AE to learn multi-scale point cloud encoding for object detection and its potential to benefit a wider range of 3D applications.

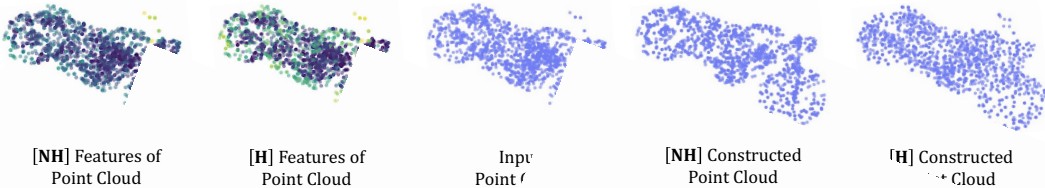

| [NH] Features of Point Cloud | [H] Features of Point Cloud | Input Point Cloud | [NH] Constructed Point Cloud | [H] Constructed Point Cloud |

Figure 4: **Visualization of fine-grained information.** We denote the outputs from hierarchical and non-hierarchical architectures as **[NH]** and **[H]**, respectively. For an input point cloud (Middle), we visualize its extracted features (Left) and reconstruction results (Right).

Table 7: **Hierarchical Modules.** 'H' represents the encoder and decoder with multi-stage hierarchies. 'Skip C.' denotes the skip connections.

| Encoder | Decoder | Skip C. | Acc. (%) |
|---------|---------|---------|----------|
| H | H | ✓ | **92.9** |
| - | - | ✓ | 90.7 |
| - | H | ✓ | 91.5 |
| H | - | ✓ | 92.2 |
| H | H | - | 92.1 |

Table 8: **Different Masking Strategy.** 'MS Mask' and 'Ratio' denote the multi-scale masking and the mask ratio.

| MS Mask | Ratio | Acc. (%) |
|---------|-------|----------|
| ✓ | 0.8 | **92.9** |
| - | 0.8 | 88.4 |
| ✓ | 0.6 | 92.3 |
| ✓ | 0.7 | 92.7 |
| ✓ | 0.9 | 92.5 |

## 4.3 Ablation Study

We conduct ablation study by modifying one of the components at a time during pre-training and explore the best masking strategy. We report the classification accuracy on ModelNet40 [44] by linear SVM to evaluate the pre-trained representations. For downstream tasks, we train the network from scratch to validate the significance of our hierarchical pre-training.

**Hierarchical Modules.** As reported in Table 7, on top of our final solution, Point-M2AE, in the first row, we respectively experiment with removing the hierarchical encoder, hierarchical decoder, and skip connections from our framework. Specifically, we replace our encoder and decoder with 1-stage plain architectures similar to MAE, which contains 15 and 2 vanilla transformer blocks, respectively. We observe the absence of multi-stage structures either in encoder or decoder hurts the performance, and the hierarchical encoder plays a better role than the decoder. Also, the skip connections well benefits the accuracy by providing complementary information for the decoder.

**Masking Strategy.** In Table 8, we report Point-M2AE with different mask settings. Without the multi-scale masking, we randomly generate masks at each scale, which leads to fragmented visible regions for all scales. With this strategy, the network would 'peek' different parts of the point cloud at different stages, which disturbs the representation learning and harms the performance by -4.5% accuracy. For different mask ratios, we find the 80% ratio performs the best to build a properly challenging pretext task for self-supervised pre-training.

**With and Without Pre-training.** We report the performance of Point-M2AE on downstream tasks with and without the pre-training in Table 9. For 'w/o', we randomly initialize the parameters and train the network from scratch. As shown, the pre-training can largely boost the performance on four datasets respectively by +1.5%, +2.5%, +3.8%, and +1.1%, which indicates the superiority and significance of our hierarchical pre-training.

Table 9: **With and without the pre-training.** 'ModelNet40-FS' denotes the few-shot classification on 10-way 20-shot ModelNet40 [44].

| Dataset | w/o (%) | w (%) |
|---------|---------|-------|
| ModelNet40 [44] | 92.5 | 94.0 |
| ScanObjectNN [38] | 83.9 | 86.4 |
| ModelNet40-FS [44] | 91.2 | 95.0 |
| ShapeNetPart [48] | 85.4 | 86.5 |

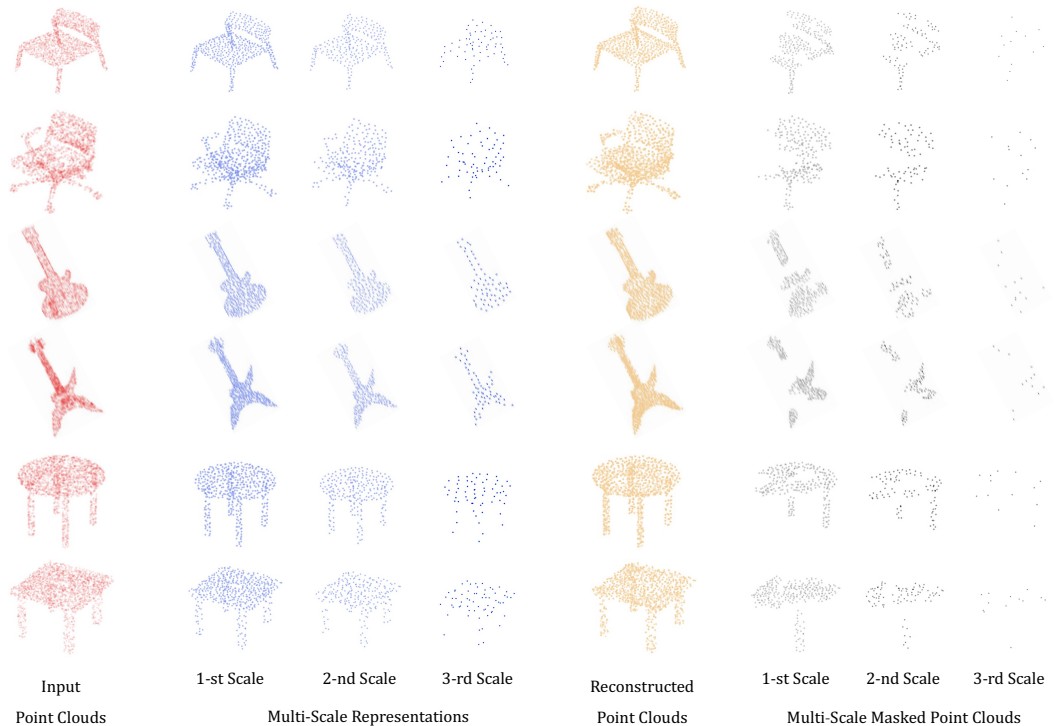

| Input Point Clouds | 1-st Scale | 2-nd Scale | 3-rd Scale | Reconstructed Point Clouds | 1-st Scale | 2-nd Scale | 3-rd Scale |
| | Multi-Scale Representations | | | | Multi-Scale Masked Point Clouds | | |

Figure 5: **Visualization of multi-scale point clouds.** In each row, we visualize the input point clouds, their multi-scale representations, the reconstructed coordinates, and multi-scale masked point clouds.

## 5 Visualization

**Multi-scale Masking.** To ease the understanding of our multi-scale masking strategy, we visualize the input point cloud, the 3-scale representations, the reconstructed point cloud, and 3-scale masked point clouds, respectively in each row of Figure 5. As shown, different scales can represent different levels of geometric details and semantics for point clouds. By the multi-scale masking strategy, we observe the visible positions of masked point clouds are block-wise within one scale and consistent across scales, which is significant for our hierarchical pre-training.

**Fine-grained Information.** The fine-grained 3D structures, e.g., thin branches of a plant, fingers of a human, engines of a plane, are significant to distinguish similar shapes and can be well encoded by our hierarchical representations. In Figure 4, we compare our Point-M2AE with multi-stage, [H], and single-scale, [NH], architectures by visualizing their extracted point features and reconstructed point clouds during pre-training. In contarst to the single-scale network, the multi-scale one indicates higher feature responses in the fine-grained structures and reconstructs more accurate spatial details.

## 6 Conclusion

We propose Point-M2AE, a multi-scale masked autoencoder for self-supervised pre-training on 3D point clouds. With a hierarchical architecture, Point-M2AE learns to produce powerful 3D representations by encoding multi-scale point clouds and reconstructing the masked coordinates from a global-to-local upsampling scheme. Extensive experiments have demonstrated the superiority of Point-M2AE to be a strong 3D representation learner. For limitations and future work, we will focus on applying Point-M2AE for wider 3D applications, e.g., outdoor and open-world scene understanding. We do not foresee negative social impact from the proposed work.

**Acknowledgement.** This work is supported by the National Natural Science Foundation of China (Grant No. 62206272), Shanghai Committee of Science and Technology (Grant No. 21DZ1100100), Centre for Perceptual and Interactive Intelligence Limited, and the General Research Fund through the Research Grants Council of Hong Kong (Grant No. 14204021, 14207319).

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
