# OpenReview forum: "Point-M2AE: Multi-scale Masked Autoencoders for Hierarchical Point Cloud Pre-training"
_NeurIPS.cc/2022/Conference — NeurIPS 2022 Accept_

### Official Review · Reviewer_3boA · 2022-07-04

**Rating:** 6
**Confidence:** 4
**Soundness:** 3 good
**Presentation:** 3 good
**Contribution:** 2 fair

**Summary:**

This paper extends the MAE to deal with irregular 3d point clouds. It proposes a multi-scale u-net like encoder-decoder to learn both fine-grained local shape information and global semantic information. The network is self-supervised trained by masking the input point cloud with multi-scale mask strategy. Experiments are conducted on ModelNet40 with the learned representation combined with SVM, achieving good classification results which is comparable to the SOTA supervised methods. The effectiveness of the proposed method is also validated by fine-tuning on the down-stream tasks such as classification, part segmentation and 3d object detection, and observed improvements over existing methods.

**Questions:**

please see the above negative points.

**Limitations:**

yes

**Strengths And Weaknesses:**

Positives
+: The paper studies an important problem in deep learning and computer vision community. The proposed method is a reasonable extension of MAE to deal with point clouds.
+: Good results were obtained by the proposed method on different tasks.
+: Detailed ablation study to validate the effectiveness of each component.

Negatives
-: It would be better if the authors can also provide the results of using the frozen encoder for other tasks like part segmentation and 3d object detection, so as to show that the conclusion about the learned representation is general enough.
-: As the advantage of self-supervised learning is to harvest from a large amount of unlabeled data, the paper lacks results about using more dataset for pre-training, for instance, using all three kinds of datasets (ShapeNet, ModelNet, ScanNet) together. This is not only helpful for readers to know the potential of the proposed method, but also can be interested to see if the proposed method can deal with different kinds of data together to boost the performance.
-: The influence of different amount of pre-training examples should also be included in the ablation study.
-: For the paper writing, it is unclear about the difficulties of introducing the MAE into 3d point cloud. It seems that there are MAE methods for 2D images, then why not we use it for 3d point, and so, the paper makes it done. Therefore, the contribution of this paper is not significant.

---

> ### Author Response · Authors · 2022-08-02
> **Response to Reviewer 3boA (Part 2/2)**
>
> >**Q3: The influence of different amount of pre-training data.**
>
> To verify the importance of training data amount, we fix to use only synthetic shapes and sample 20\%, 40\%, 60\%, and 80\% of ShapeNet data for pre-training. We also adopt the linear SVM on ModelNet40 and fine-tuning on ScanObjectNN as the evaluation metrics. As reported in the table below, more pre-training data contributes to better downstream performance, which accords with the intuition.
>
>
> |Data Amount |0\% |20\% |40\% |60\% |80\% |100\%|
> |--|-|-|-|-|-|-|
> |ModelNet40 |0 |89.7 |90.8 |91.6 |92.1 |**92.9**|
> |ScanObjectNN |83.9 |84.2 |85.1 |84.9 |85.7 |**86.4**|
>
>
> >**Q4: Difficulties of introducing MAE into 3D point clouds.**
>
> Thanks for the suggestion. We will improve the introduction part in the revised paper. There are four main difficulties of directly transferring MAE from 2D images to 3D point clouds.
>
> >>**1. The irregular data form of point clouds.**
>
> 2D images are grid-based data, whose pixels have spatially regular arrangements. By this, 2D MAE can naively divide the image into non-overlapping patches and randomly mask some of them for reconstruction. In contrast, 3D point clouds are permutation-invariant and are irregularly distributed in 3D space. How to convert point clouds into multiple discrete tokens that can be masked and reconstructed is an important challenge to be tackled. For this, we utilize the widely adopted Farthest Point Sampling (FPS) to obtain the token centers and adopt $k$-NN to aggregate neighboring features as the token features. The FPS makes the point tokens evenly scatter in the space and has minimum overlaps, which prevents information leakage between masked and unmasked tokens. Also, the $k$-NN ensures that each masked token only requires to reconstruct its neighboring points, creating a properly challenging pretext task.
>
> >>**2. The local-to-global relations of 3D structures.**
>
> Unlike 2D images, it is critical to understand the relations between local parts and the overall 3D shape, which have geometric and semantic dependence. For example, the network can recognize an airplane starting from its wings, or segment the wing's part from the airplane's global feature.
> However, 2D MAE directly downsamples the image into a low-resolution feature map and adopt a non-hierarchical transformer to process. Therefore, we propose a hierarchical MAE architecture unique for 3D point clouds. Our Point-M2AE has multiple stages that progressively encodes different point cloud scales, thus better encoding the local-to-global relations.
>
> >>**3. How to mask multi-scale point cloud?**
>
> As 2D MAE only has a single image scale, it only needs a random mask over the transformer. For our Point-M2AE, as illustrated in the main paper, we are required to generate multi-scale masks that ensure the visible regions to be consistent across scales. This is for preserving complete local patterns and enabling coherent learning of the encoder. Otherwise, the detailed 3D geometry would be lost and the encoder would `see' different unmasked parts of the point clouds at different scales, which severely harms the performance (92.9\% $\rightarrow$ 88.4\% of Linear SVM on ModelNet40).
>
> >>**4. How to capture fine-grained 3D structures?**
>
> The fine-grained information of 3D structures are significant for downstream 3D dense prediction tasks, e.g., part segmentation. Except for the multi-scale architecture and masking, we further add skip connections between the encoder and decoder, which has not been tried before on 2D MAE. This can complement the fine-grained point cloud features to the decoder and improve the performance of Point-M2AE.
>
> In summary, our Point-M2AE considers challenges that are distinct for 3D representation learning, and introduces specific designs accordingly.

---

> ### Author Response · Authors · 2022-08-02
> **Response to Reviewer 3boA (Part 1/2)**
>
> We sincerely thank your helpful suggestions, and address the concerns as follows:
>
> >**Q1: Frozen encoder for other tasks.**
>
> Thank you for the suggestion. Prior works only test the frozen encoder by training a linear SVM on ModelNet40 for synthetic shape classification, and it is more reasonable to show the learned representation for other 3D tasks. We experiment our Point-M2AE and Point-BERT with their frozen encoders on three other downstream datasets: real-world shape classification on ScanObjectNN, part segmentation on ShapeNetPart, and few-shot classification (5-way 20-shot) on ModelNet40. For real-world and few-shot classification, we append a learnable classification head of linear projection layers to the pre-trained encoder. For part segmentation, we make the segmentation decoders of both Point-M2AE and Point-BERT unfrozen, whose architectures are the same as the fully-unfrozen fine-tuning experiments. As Point-BERT does not provide the pre-training approach or weights for 3D object detection, we cannot compare their detection performances here.
>
> The results are presented in the following table. With the frozen encoder, Point-M2AE consistently outperforms Point-BERT on all tasks, e.g., +5.5\% on ModelNet40 and +2\% on 5-way 20-shot classification. With the 85.6\% mIoU_I, Point-M2AE's frozen encoder even performs comparably to the fully fine-tuned Point-BERT. The results fully demonstrate our pre-training has learned better and more general point cloud representations than Point-BERT does.
>
> ||Frozen Encoder|ModelNet40 |ScanObjectNN |ShapeNetPart |5-way 20-shot|
> |---|---|---|---|---|---|
> |Point-BERT |Yes |87.4  |75.6 |84.8 |97.0 ± 2.3|
> |Point-M2AE |Yes |**92.9**|**78.3**|**85.6**|**97.2 ± 2.1**|
> |||+5.5\% |+2.7\% |+0.8\% |+0.2\%|
> |Point-BERT |No |93.2  |83.1 |85.6 |96.3|
> |Point-M2AE |No |**94.0**|**86.4**|**86.5**|**98.3**|
> |||+0.8\% |+2.7\% |+0.9\% |+2.0\%|
>
> >**Q2: More unlabeled data for pre-training.**
>
> Thank you for the suggestion. The great advantage of unsupervised learning is to utilize large-scale unlabeled data. We first categorize the current available datasets according to the types of point clouds below.
>
>
> |Datasets|ShapeNet |ModelNet40 |ScanObjectNN |ScanNetV2|
> |---|---|---|---|---|
> |Point Clouds |Synthetic shapes |Synthetic shapes |Real-world shapes  |Real-world Scenes|
> |Training Samples |57,448 |9,843 |11,416 |1,201|
>
>
> The default setting of prior works is to pre-train on the synthetic ShapeNet and fine-tune on the others. We here incorporate more unlabeled data for pre-training Point-M2AE and present the results in the table below.
>
> |ShapeNet |ModelNet40 |ScanObjectNN |ScanNetV2 |ModelNet40 |ScanObjectNN|
> |---|---|--|--|--|--|
> |✔️ |- |- |- |92.9 |86.4|
> |✔️ |✔️ |- |- |**93.1**|86.5|
> |✔️ |- |✔️ |- |92.3 |87.1|
> |✔️ |- |- |✔️ |91.2 |86.8|
> |✔️ |✔️ |✔️ |✔️ |92.6 |**87.6**|
>
> Considering different types of point clouds, We adopt two evaluation metrics, classification with a linear SVM on ModelNet40, and fine-tuning on ScanObjectNN, which respectively reflect the pre-trained representations for synthetic and real-world point clouds. As shown in the table, with more pre-training point clouds of the same type, the downstream performance can be largely improved.
>
> 1) If the synthetic ModelNet40 is integrated with synthetic ShapeNet, the classification accuracy of the linear SVM test on ModelNet40 is boosted to 93.1\%. Also, pre-trained by more 3D-shape point clouds, the classification accuracy of fine-tuning on ScanObjectNN with real-world shapes can be slightly improved.
>
> 2) If we incorporate the real-world ScanObjectNN or ScanNetV2 during pre-training, the ModelNet40 scores would be slightly harmed due to the domain gap, but the ScanObjectNN scores are both boosted for training more real-world data.
>
> 3) If all datasets are utilized as pre-training data, the classification accuracy of fine-tuning on ScanObjectNN can be largely improved to 87.6\%. This demonstrates the learning capability of Point-M2AE if more pre-training data is available.

---

### Official Review · Reviewer_Ddan · 2022-07-10

**Rating:** 5
**Confidence:** 4
**Soundness:** 3 good
**Presentation:** 3 good
**Contribution:** 2 fair

**Summary:**

This work proposes a multi-scale masked autoencoder for pre-training on point clouds. It designs a pyramid architecture for hierarchical encoder and decoder to progressively model 3D point clouds. A novel multi-scale masking and a back-project strategy are proposed for generating consistent visible regions across scales for point clouds. Extensive experiments demonstrate that the proposed method outperforms other pre-training methods on the linear SVM classification task and downstream tasks.

**Questions:**

1. Why does the final experiment use only 2-3 scales and what about more? Please explain further the motivation for the choice of the number of scales.
2. In Sec 3.2 ‘Token Embedding and Merging’, “After that, we obtain the initial point tokens T_1^v for the 1-st stage of the encoder” seems to be conflict with the Figure 2 where the T_1^v is the output of the 1-st stage of the encoder.
3. In Sec 4.2 ‘Shape Classification’, “We follow Point-BERT to use the voting strategy [29] for fair comparison on ModelNet40…”, as the Point-BERT does not seem to mention that it uses a voting strategy and dose not cite the paper [29]. Please explain further the voting strategy.



**Ethics Review Area:**

["I don’t know"]

**Strengths And Weaknesses:**

Strengths:
1. This paper is generally technically sound.
2. The design of the masking and back-project strategy is novel and fits well on the 3D point clouds pretraining task.
3. The extensive experiments verify the effectiveness of the proposed approach.

Weakness:
1. The overall framework seems to be simply a variant of PoinNet++ with Transformer Encoder, especially the idea of the Skip Connection and Point Token Upsampling, hence the framework seems to have insufficient technical innovations.
2. It can be noted that the U-Net like network demonstrated in Figures 2 and 3 have arbitrary multiple scales (at least greater than three). However, the actual experiments have only 2-3 scales and do not give an adequate explanation for the choice of the number of scales.

---

> ### Author Response · Authors · 2022-08-02
> **Response to Reviewer Ddan (Part 2/2)**
>
>
>
> >**Q2: Explanation of the scale number.**
>
> Thank you for the suggestion. We will add more detailed explanations for the number of scales in the revised paper. We have conducted ablation study of different scale numbers in Table 2 of the original supplementary material, and we also list the results in the following tables. Here are two points about the choice of scale number:
>
> >>**1. Scale number depends on the input point number.**
>
> Our multi-scale masking back-projects the visible regions by the neighbor indices, which actually have some overlapping for the k-NN of neighboring points. If the input number is fixed, too many scales would accumulate to more visible regions and finally cause most tokens to be visible at the lowest 1-st scale. Then, the reconstruction pretext task becomes less challenging and will quickly converge, by which the encoder cannot learn robust point cloud representation. In the paper, we adopt 2,048 input points for pre-training, and the best scale number is 3.
> If we adopt 4-scale Point-M2AE, we report the visible point numbers of different scales in the following table. As shown, even if the mask ratio is 80\% at the highest 4-th scale (6/32 visible), the back-projected 2-nd scale (164/256 visible) and 1-st scale (458/512 visible) nearly have no masked points.
>
> | |1-st scale |2-nd scale |3-rd scale |4-th scale|
> |---|---|---|---|---|
> |All Tokens |512 |256 |64 |32|
> |Visible Tokens |458 |164 |27 |6|
>
>
> We then experiment with 1,024 and 4,096 input points in the following tables, where more points can perform better with more scales.
>
> |Scale |Points |Acc (\%)|
> |---|---|---|
> |**2** |**1024** |**92.3**|
> |3 |1024 |91.9|
> |**3** |**2048**|**92.9**|
> |4 |2048 |90.4|
> |4 |4096 |92.8|
> |**5** |**4096** |**93.1**|
>
>
> >>**2. The stage number of decoder is better to be one fewer than the scale number.**
>
> Suppose we build an S-scale Point-M2AE, the encoder and decoder are better to be $S$-stage and (S-1)-stage, respectively. As explained in Section 3.3 of the main paper, the (S-1)-stage of the decoder (corresponds to the 2-nd scale of the point cloud) can already well represent the overall 3D shape and simultaneously preserve enough local patterns, referring to the visualization in Figure 3's $P_2$ of the paper. If reconstructing from this (S-1)-stage, the pretext task can be more challenging without fine-grained cues of $P_1$. Further upsampling point clouds into $1$-st scale at the decoder's S-th stage would only bring extra computation budget and harm the representation learning by a too simple pretext task. We experiment with equal stage numbers of encoder and decoder in the table below.
>
> |Encoder |Decoder |GPU Mem. |Acc (\%)|
> |---|---|---|---|
> |**3** |**2** |**19 GiB** |**92.9**|
> |3 |3 |24 GiB |90.7|
>
>
>
> >**Q3: The notation conflict of $T_1^v$.**
>
> Sorry for the misunderstanding. For simplicity, we abuse the notation of the point tokens of the s-th stage as $T_s^v$, regardless whether they are before or after the s-th stage encoder, since the $s$-th stage encoder would not change the scales of tokens.
>
> >**Q4: The voting strategy of Point-BERT.**
>
> The paper of Point-BERT indeed does not mention the voting strategy, but marks this point in its official GitHub repository. In the "Pretrained Models" section of the README, we can observe "Acc. (vote)" at the second table's header. It shows that Point-BERT's results on ModelNet40 of 93.24 (1k points), 93.48 (4k points), and 93.76 (8k points) are all obtained based on voting. More specifically, the code for voting is at line 382 at ***Point-BERT/tools/runner_BERT_finetune.py*** of its repository. Therefore, we also report our classification accuracy, 94.0\% with 1k points, on ModelNet40 by the same voting strategy as Point-BERT for fair comparison.
>
> To conduct voting, the trained model is utilized to predict the test set for 10 times by default. In each time, the test point clouds are randomly transformed by scaling and translation. After that, the classification logits for each test point cloud are integrated by max pooling, which can increase the model's inference robustness.

---

> ### Author Response · Authors · 2022-08-02
> **Response to Reviewer Ddan (Part 1/2)**
>
> We sincerely thank your constructive advice, and address the reviewer's concerns as follows:
>
> >**Q1: The framework compared to PointNet++.**
>
> As PointNet++ laid the groundwork for hierarchical point cloud processing, nearly all later point-based works inherited its multi-scale framework and inserted more advanced geometry extractors on top of it, such as DGCNN, CurveNet, PointMLP, etc. However, our Point-M2AE extends the multi-scale architecture for a totally different task, point cloud masked autoencoding. The distinct differences are as follows:
>
> >>**1. A new task.**
>
> Although PointNet++ has been explored for 3D multi-scale feature extraction, for masked autoencoding on point cloud pre-training, we are the first to successfully learn representations with a multi-scale architecture. Even for masked autoencoding on 2D images, there is no prior work to adopt a multi-scale transformer or an encoder-decoder with skip connections prior to NeurIPS's submission deadline.
>
>
> >>**2. Architecture differences.**
>
> Directly using PointNet++-like hierarchical transformer cannot achieve competitive performance as shown in the ablation study of the main paper (Tables 7 and 8). Only with our proposed modifications, the classification accuracy of ModelNet40 dataset with a Linear SVM can be boosted from 88.4\% to 92.9\%.
> The modifications are as follows:
>
> 1) We propose the multi-scale masking and the mask-guided token merging/propagating modules. As our mask is back-projected across scales via neighbor indices, the merging/propagating process are required to be unified with the indices. Otherwise, it would cause inconsistent visible regions across scales and information leakage between masked and unmasked tokens.
>
> 2) During self-supervised pre-training, we utilize the hierarchical decoder with skip connections and token upsampling. For fine-tuning on part segmentation task, we design a new decoder as illustrated in Section 4.2 of the paper. It contains no skip connections or hierarchical upsampling, but directly upsamples multi-scale features into 2,048 points, which is different from PointNet++'s decoder.
>
> 3) The skip connections are only utilized to supplement fine-grained features for the ***visible tokens*** in the decoder, since our encoder only processes the unmasked tokens. In contrast, PointNet++ processes all input points and constructs the connections from every pair between the encoder and decoder.
>
> >>**3. Different implementation details.**
>
> Besides the modifications above, we also have different implementation details:
>
> 1) PointNet++ adopts mini-PointNet to aggregate local point features, but our Point-M2AE utilizes transformers with local spatial self-attention mechanisms.
>
> 2) PointNet++ uses ball query to search neighboring points for downsampling, but we utilize k-NN algorithm.
>
> 3) Our decoder has one fewer stage than the encoder to build a properly challenging pretext task for pre-training, but PointNet++ has the same stage numbers for the encoder and decoder.

---

### Official Review · Reviewer_774k · 2022-07-11

**Rating:** 6
**Confidence:** 4
**Soundness:** 3 good
**Presentation:** 4 excellent
**Contribution:** 3 good

**Summary:**

This paper uses MAE to learn the point cloud features without fixed topology. Its multi-scale architecture can capture features at different resolutions for guidance. The pyramid architecture also facilitates the network to extract high-level features, and the self-attention mechanism adopted by this network also controls the focus of the network. The complementary skip connections also better connect the global and local features to guide each other. And it also produces competitive results in experiments.

**Questions:**

1) There are some works in the field that apply masked autoencoders in point clouds except Point-BERT, but I find that the authors miss these works[1][2][3][4], and I want to see more discussion from the authors about some of these out-picked works, and what this paper amounts to in terms of improvements to these precursor works.
2) I hope that the authors can add visual interpretation of the results of the Local Spatial Self-Attention module in their ablation experiments.
3) The authors keeps stressing that their model can pay more attention to fine-grained information, but there are no more descriptions about fine-grained information except \textbf{Figure 2} in \textbf{Appendix C} (which only shows the point cloud images with different scales).
4) Although the authors have achieved good results in the experiments after freezing the encoder for the relevant downstream tasks, have you analyzed the influence of the CD loss you selected for the training of the model's results? Now it seems that the decoder is driven only by CD Loss. Whether to use EMD as loss in the experiments or combine the two as compound loss? Will the results still be as good as now? I'd like to see the quantitative results if possible.
5) Miss the figure tag in \textbf{Appendix C. Line 68}: “As shown in Figure []”.

In the current version, there are some issues as well. I look forward to the response by the authors. For now, I would recommend a borderline accept rating for the paper.

[1]Fu, Kexue, et al. "POS-BERT: Point Cloud One-Stage BERT Pre-Training." arXiv preprint arXiv:2204.00989 (2022).

[2]Liu, Haotian, Mu Cai, and Yong Jae Lee. "Masked Discrimination for Self-Supervised Learning on Point Clouds." arXiv preprint arXiv:2203.11183 (2022).

[3]Pang, Yatian, et al. "Masked autoencoders for point cloud self-supervised learning." arXiv preprint arXiv:2203.06604 (2022).

[4]Pellis, Eugenio, et al. "An Image-Based Deep Learning Workflow for 3D Heritage Point Cloud Semantic Segmentation." International Archives of the Photogrammetry, Remote Sensing and Spatial Information Sciences-ISPRS Archives 46.2/W1-2022 (2022): 429-434.

**Limitations:**

Yes, I don't see any potential negative social impact of this work.

**Strengths And Weaknesses:**

- Originality: The main idea of the proposed approach is to use the masked autoencoders. To my limited knowledge about it, this is novel and thus the originality is reasonable.
- Quality：While the approach seems reasonable and the experimental results look promising, I have the following concerns(See Questions) about the paper.
- Clarity：This manuscript is clearly written.
- Significance：There are not many works in the field of point cloud feature learning by masked autoencoders, so I think this paper makes a positive contribution.

---

> ### Author Response · Authors · 2022-08-02
> **Response to Reviewer 774k (Part 2/2)**
>
> >**Q3:  More description of fine-grained information.**
>
> Thanks for the valuable suggestion. The fine-grained information of the point clouds refers to the exquisite 3D structures and subtle geometric variations, e.g., thin branches of a plant, fingers of a human, engines of a plane, etc. It can be better revealed by high-resolution points and are more significant to some dense 3D tasks, e.g., part segmentation. However, the plain transformer of vanilla MAE is a non-hierarchical architecture and directly downsamples the input point cloud into low-resolution cubes, which would severely blur such details.
>
> Therefore, our Point-M2AE adopts a pyramid encoder-decoder transformer with multi-scale representations. In this way, the network can well capture the fine-grained features from the reserved fine-grained 3D structures in the early stages and gradually aggregates them into high-level semantics.
>
> To better ease the understanding, we show the differences between multi-scale (hierarchical) and single-scale (non-hierarchical) architectures by two visualizations in ***the newly-revised supplementary material***. In ***Figure 6***, we visualize the extracted point features and the reconstructed masked point clouds during pre-training. Compared to the non-hierarchical network, the hierarchical one shows higher feature responses in the fine-grained structures and reconstructs details more accurately. In ***Figure 7***, we visualize the extracted point features and the segmentation results in downstream part segmentation task. Likewise, the multi-scale architecture predicts more fine-grained part labels for the objects.
>
> >**Q4: EMD Loss.**
>
> Thanks for this suggestion. Except for the Chamfer Distance (CD) loss with L2 norm, we further evaluate the L1-norm CD loss, EMD loss, and their combinations with a Linear SVM on ModelNet40. As shown in the table below, the original L2-norm loss performs better than all other compared losses.
>
>
> |L2-norm CD |L1-norm CD |EMD |Linear SVM|
> |---|---|---|---|
> |✔️ |- |-  |**92.9**|
> |- |✔️ |-  |91.1|
> |- |- |✔️  |91.9|
> |✔️|- |✔️  |92.4|
> |- |✔️ |✔️ |91.3|
>
> We denote the reconstructed and ground-truth point sets as $S_1$ and $S_2$. Compared to EMD loss that requires an optimal mapping for every point between $S_1$ and $S_2$, L2-norm CD loss only optimizes the separate pair-wise distances and is thus more robust to the variation of 3D structures. Compared to L1-norm CD loss, L2 norm of Euclidean Distances can better depict spatial distribution and pay more attention to the far away points.
>
> >**Q5: Miss figure tag.**
>
> Thanks for pointing out. We will correct the tag in the revised supplementary material.

---

> ### Author Response · Authors · 2022-08-02
> **Response to Reviewer 774k (Part 1/2)**
>
> We sincerely thank your insightful comments, and address the concerns as follows:
> >**Q1: Comparison to similar works with masked modeling on point clouds.**
>
> Thanks for mentioning the related work. We will cite and discuss them in our revised paper.
>
> Fu et al. [1], Liu et al. [2], and Pang et al. [3] also conduct point cloud pre-training via masking, which are ***concurrent works*** to ours, but use different strategies for masked modeling.
>
> >>**Comparison to Fu et al [1]:**
>
> 1) **Different pre-training strategies.** Following Point-BERT, [1] utilizes BERT-style pre-training. It is not a masked autoencoder (MAE) and different from our MAE-style pre-training. Such BERT style predicts the masked token encoded by an independently trained tokenizer, while our MAE style directly reconstructs the masked points' raw 3D coordinates, which is simpler and more efficient.
>
> 2) **Less self-supervisory signals.** [1] consists of two complicated losses, a masked modeling loss and a contrastive loss for different sub-sets of point clouds. Our Point-M2AE only requires the simple reconstruction loss and achieves better performances.
>
> >>**Comparison to Liu et al [2]:**
>
> 1) **Different pre-training strategies.** [2] proposed a masked discrimination (MD) pre-text task that conducts binary classification to judge if a point token is masked. It adopts binary focal loss for self-supervision and is different from our MAE-style pre-training that reconstructs masked coordinates.
>
> >>**Comparison to Pang et al [3]:**
>
> 1) **Hierarchical architectures.** [3] also adopts MAE-style pre-training but utilizes a plain transformer-like 2D MAE without 3D specific modifications. Our Point-M2AE adopts a hierarchical encoder-decoder with skip connections and local attention to better capture local-to-global 3D geometries.
>
> 2) **Multi-scale Masking strategy.** [3] adopts the vanilla random masking, but we introduce a multi-scale masking to generate consistent visible region across scales. It can largely boosts the performance as shown in Table 7 of the main paper (88.4 $\rightarrow$ 92.9 for Linear SVM on ModelNet40).
>
> >>**Comparison to Pellis et al [4]:**
> 1) **Different tasks.** We target on the self-supervised point cloud pre-training, but [4] solves semantic segmentation specially for heritage point clouds without pre-trained 3D networks. Thus, [4] cannot be compared with our method for pre-training.
>
> 2) **Different usage of masks.** [4] leverages the masks to project labeled multi-view images into point clouds, while we utilize masks for pre-training via masked autoencoding.
>
> We compare the characteristics and performances of [1], [2], [3] on different tasks in the following table.
> 'Linear SVM' denotes the linear evaluation on ModelNet by SVM, and '5-way 20-shot' denotes the few-shot classification on 5-way 20-shot ModelNet40. The last three lines represent the three fine-tuning experiments on three datasets.
>
> ||Point-M2AE|Point-BERT|[1]|[2]|[3]|
> |---|---|---|---|---|---|
> |Pre-training Style|MAE|BERT|BERT |MD |MAE|
> |Hierarchical |Yes |No |No  |No |No|
> |Attention Scope |Local |Global |Global  |Global |Global|
> |Masking |Multi-scale |Random |Random  |Random |Random|
> |Linear SVM |**92.9**| 87.4 |92.1  |- |-|
> |5-way 20-shot |**98.3** |96.3 |97.0 |97.2 |97.8|
> |ModelNet40 |**94.0**|93.2 |93.6 |93.8 |93.8|
> |ScanObjectNN |**86.4**|83.1 |83.2 |84.3 |85.2|
> |ShapeNetPart |**86.5**|85.6 |86.0 |86.0 |86.1|
>
> >**Q2: Visual interpretation of local spatial attention in ablation study.**
>
> We visualize the attention weights with and without the local attention in ***Figure 5 of the newly-revised supplementary material.*** As shown in the figure, with the local attention, the query point (marked by star) only has large attention values within a local spatial range (marked by yellow dotted circles), other than scattering over the entire 3D shape (marked by yellow arrows). This enables each point to concentrate more on neighboring local features in early stages for capturing and encoding detailed structures.
>
> References
>
> [1] POS-BERT: Point Cloud One-Stage BERT Pre-Training. arXiv 2022.
>
> [2] Masked Discrimination for Self-Supervised Learning on Point Clouds. arXiv 2022.
>
> [3] Masked Autoencoders for Point Cloud Self-supervised Learning. ECCV 2022.
>
> [4] An Image-Based Deep Learning Workflow for 3D Heritage Point Cloud Semantic Segmentation. International Archives of the Photogrammetry, Remote Sensing and Spatial Information Sciences-ISPRS Archives 46.2/W1-2022 (2022): 429-434.

---

### Author Response · Authors · 2022-08-08
**Looking forward to further discussion**

Dear Reviewers,

Thanks again for your insightful comments and valuable time in reviewing our paper. We have provided the corresponding responses to your concerns, and added them in ***the revised supplementary material*** accordingly, which are highlighted in blue.

Given the discussion phase is quickly passing, we wonder if our responses address your concerns. If you have any further questions, we are more than happy to discuss them.

Looking forward to your reply.

Best, Authors

---

### Meta-Review · Area_Chair_X9YT · 2022-08-21

**Recommendation:** Accept
**Confidence:** Certain

**Metareview:**

This paper proposes, the Point-M2AE, a multi-scale masked autoencoder (MAE) pre-training framework for self-supervised learning of 3D point clouds. This is a generalization of the existing 2D-MAE framework to 3D point cloud domain. The proposed Point-M2AE introduces a U-Net-like transformer and a multi-scale masking strategy to generate consistent visible regions across scales. Extensive experiments are conducted on various downstream tasks to validate the power of the proposed method.  All reviewers think that the current paper presents a novel framework, making an important contribution to point cloud representation. It also has sufficient empirical results to demonstrate the performance of the proposed model. The feedback from the authors also addresses the major concerns of the reviewers. After reading the reviewers’ comments and the authors’ replies, the AC recommends accepting the paper.

**Award:**

No

---

### Decision · Program_Chairs · 2022-09-14

Accept